# HYBED: HYPERBOLIC NEURAL GRAPH EMBEDDING

## ABSTRACT

Neural embeddings have been used with great success in Natural Language Processing (NLP) where they provide compact representations that encapsulate word similarity and attain state-of-the-art performance in a range of linguistic tasks. The success of neural embeddings has prompted significant amounts of research into applications in domains other than language. One such domain is graph-structured data, where embeddings of vertices can be learned that encapsulate vertex similarity and improve performance on tasks including edge prediction and vertex labelling. For both NLP and graph-based tasks, embeddings in high-dimensional Euclidean spaces have been learned. However, recent work has shown that the appropriate isometric space for embedding complex networks is not the flat Euclidean space, but a negatively curved hyperbolic space. We present a new concept that exploits these recent insights and propose learning neural embeddings of graphs in hyperbolic space. We provide experimental evidence that hyperbolic embeddings significantly outperform Euclidean embeddings on vertex classification tasks for several real-world public datasets.

## 1 INTRODUCTION

Embeddings are used to represent complex high-dimensional data in lower-dimensional continuous spaces (Roweis & Saul, 2000; Belkin & Niyogi, 2001). Embedded representations provide three principal benefits over sparse schemes: They encapsulate similarity, are compact, and perform better as inputs to machine learning models (Salton et al., 1975). These benefits are particularly important for graph-structured data where the native representation is the adjacency matrix, which is typically a sparse matrix of connection weights.

Neural embedding models are a flavour of embedding where the embedded representation corresponds to a subset of the connection weights in a neural network (see Fig. 2a), which are learned through backpropagation. Neural embedding models have been shown to improve performance on many tasks across multiple domains, including word analogies (Mikolov et al., 2013a; Mnih, 2013), machine translation (Sutskever et al., 2014), document comparison (Kusner et al., 2015), missing edge prediction (Grover & Leskovec, 2016), vertex attribution (Perozzi & Skiena, 2014), product recommendations (Grbovic et al., 2015; Baeza-Yates & Saez-Trumper, 2015), customer value prediction (Kooti et al., 2017; Chamberlain et al., 2017) and item categorisation (Barkan & Koenigstein, 2016). In all cases, the embeddings are learned without labels (unsupervised) from a sequence of tokens. Previous work on neural embedding models has either either explicitly or implicitly (by using the Euclidean dot product) assumed that the embedding space is Euclidean. However, recent work in the field of complex networks has found that many interesting networks, particularly those with a scale-free structure such as the Internet (Shavitt et al., 2008; Boguna et al., 2010) or academic citations (Clough et al., 2015; Clough & Evans, 2016) can be well described with a geometry which is non-Euclidean, such as hyperbolic geometry. Even more recently the problem of mapping graphs and datasets to a low-dimensional hyperbolic space has been addressed in (Nickel & Douwe, 2017) and (Bläsius et al., 2016). Here we use a neural embedding approach based on the Skipgram architecture to find hyperbolic embeddings.

There are two reasons why embedding complex networks in hyperbolic geometry can be expected to perform better than Euclidean geometry. The first is that complex networks exhibit a hierarchical structure. Hyperbolic geometry provides a continuous analogue of tree-like graphs, and even infinite trees have nearly isometric embeddings in hyperbolic space (Gromov, 2007). The second property is that complex networks have power-law degree distributions, resulting in high-degree hub vertices.

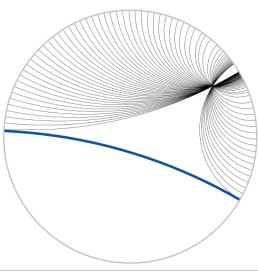 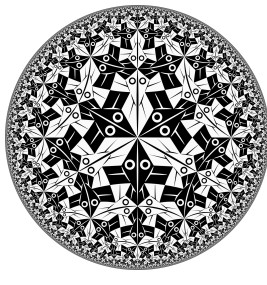 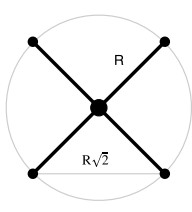

(a) Parallel hyperbolic lines.  (b) "Circle Limit 1", Escher  (c) A hub and spoke graph.

Figure 1: Properties of hyperbolic space. a Multiple parallel lines passing through a single point. b All tiles are of constant area in hyperbolic space, but shrink to zero area at the boundary of the disk in Euclidean space. c Hub and spokes graph. It is impossible to embed this graph in two-dimensional Euclidean space and preserve the properties that (1) all spokes are the same distance from the hub, (2) all spokes are the same distance from each other, and (3) the distance between spokes along the circumference is more than twice the distance to the hub. In hyperbolic space such embeddings exist.

Fig. 1c shows a simple hub-and-spoke graph where each spoke is a distance $R$ from the hub and $2R$ from each other. For an embedding in two-dimensional Euclidean space it is impossible to reproduce this geometry for more than two spokes. However, in hyperbolic space, large numbers of spokes that satisfy these geometrical constraints can be embedded because the circumference of a circle expands exponentially rather than polynomially with the radius.

The starting point for our model is the celebrated Skipgram architecture (Mikolov et al., 2013a;b) shown in Fig. 2a. Skipgram is a shallow neural network with three layers: (1) An input projection layer that maps from a one-hot-encoded token to a distributed representation, (2) a hidden layer, and (3) an output softmax layer. Skipgram is trained on a sequence of words that is decomposed into (input word, context word)-pairs. The model uses two separate vector representations, one for the input words and another for the context words, with the input representation comprising the learned embedding. The (input word, context word)-pairs are generated by running a fixed length sliding window over a word sequence. Words are initially randomly allocated to vectors within the two vector spaces. Then, for each training word pair, the vector representations of the observed input and context words are pushed towards each other and away from all other words (see Fig. 2b). The model can be extended to network structured data using random walks to create sequences of vertices. Vertices are then treated exactly analogously to words in the NLP formulation. This was originally proposed as DeepWalk (Perozzi & Skiena, 2014). Extensions varying the nature of the random walks have been explored in LINE (Tang et al., 2015) and Node2vec (Grover & Leskovec, 2016).

**Contribution** In this paper, we introduce the new concept of neural embeddings in hyperbolic space. We formulate backpropagation in hyperbolic space and show that using the natural geometry of complex networks improves performance in vertex classification tasks across multiple networks. At the same time, Nickel & Douwe (2017) independently proposed a hyperbolic embedding algorithm that has similarities to ours. The key differences are that Nickel & Douwe (2017) try to fit the hyperbolic distance between nodes using cartesian coordinates in the Poincaré disk, whereas we use a modified cosine distance in a spherical hyperbolic coordinate system. Our approach does not require a numerical constraint to prevent points from 'falling off' the edge of the disk and becoming infinitely distant from the others.

## 2 HYPERBOLIC GEOMETRY

Hyperbolic geometry emerged through a relaxation of Euclid's fifth geometric postulate (the parallel postulate). In hyperbolic space, there is not just one, but an infinite number of parallel lines that pass through a single point. This is illustrated in Fig. 1a where every fine line is parallel to the bold, blue line, and all pass through the same point. Hyperbolic space is one of only three types of isotropic space that can be defined entirely by their curvature. The most familiar is flat Euclidean space. Space with uniform positive curvature has an *elliptic geometry* (e.g. the surface of a sphere) and space with uniform negative curvature has a *hyperbolic geometry*, which is analogous to a saddle-like surface.

Unlike Euclidean space, in hyperbolic space even infinite trees have nearly isometric embeddings, making the space well suited to model complex networks with hierarchical structure. Additionally, the defining features of complex networks, such as power-law degree distributions, strong clustering and community structure, emerge naturally when random graphs are embedded in hyperbolic space (Krioukov et al., 2010).

One of the defining characteristics of hyperbolic space is that it is in some sense *larger* than Euclidean space; the 2D hyperbolic plane cannot be isometrically embedded into Euclidean space of any dimension, unlike elliptic geometry where a 2-sphere can be embedded into 3D Euclidean space etc. The hyperbolic area of a circle or volume of a sphere grows exponentially with its radius, rather than polynomially. This property allows low-dimensional hyperbolic spaces to provide effective representations of data in ways that low-dimensional Euclidean spaces cannot. Fig. 1c shows a hub-and-spoke graph with four spokes embedded in a two-dimensional Euclidean plane so that each spoke sits on the circumference of a circle surrounding the hub. Each spoke is a distance $R$ from the hub and $2R$ from every other spoke, but in the embeddings the spokes are a distance of $R$ from the hub, but only $R\sqrt{2}$ from each other. Complex networks often have small numbers of vertices with degrees that are orders of magnitude greater than the median. These vertices approximate hubs. The distance between spokes tends to the distance along the circumference $s = \frac{2\pi R}{n}$ as the number of spokes $n$ increases, and so the shortest distance between two spokes is via the hub only when $n < \pi$. However, for embeddings in hyperbolic space, we get $n < \frac{\sinh R}{R}$, such that an infinite number of spokes can satisfy the property that they are the same distance from a hub, and yet the path that connects them via the hub is shorter than along the arc of the circle. As hyperbolic space can not be isometrically embedded in Euclidean space, there are many different representations that each conserve some geometric properties, but distort others. In this paper, we use the Poincaré disk model of hyperbolic space.

## 2.1 POINCARÉ DISK MODEL

The Poincaré disk models the infinite two-dimensional hyperbolic plane as a unit disk. For simplicity we work with the two-dimensional disk, but it is easily generalised to the $d$-dimensional Poincaré ball, where hyperbolic space is represented as a unit $d$-ball. Hyperbolic distances grow exponentially towards the edge of the disk. The boundary of the disk represents infinitely distant points as the infinite hyperbolic plane is squashed inside the finite disk. This property is illustrated in Fig. 1b where each tile is of constant area in hyperbolic space, but rapidly shrink to zero area in Euclidean space. Although volumes and distances are warped, the Poincaré disk model is *conformal*. Straight lines in hyperbolic space intersect the boundary of the disk orthogonally and appear either as diameters of the disk, or arcs of a circle. Fig. 1a shows a collection of straight hyperbolic lines in the Poincaré disk. Just as in spherical geometry, shortest paths appear curved on a flat map, hyperbolic geodesics also appear curved in the Poicaré disk. This is because it is quicker to move close to the centre of the disk, where distances are shorter, than nearer the edge. In our proposed approach, we will exploit both the conformal property and the circular symmetry of the Poincaré disk. The geometric intuition motivating our approach is that vertices embedded near the middle of the disk can have more 'near' neighbours than they could in Euclidean space, whilst vertices nearer the edge of the disk can still be very far from each other.

## 2.2 SIMILARITIES, ANGLES, AND DISTANCES

The distance metric in Poincaré disk is a function only of the radius. Exploiting the angular symmetries of the model using polar coordinates considerably simplifies the mathematical description of our approach and the efficiency of our optimiser. Points in the disk are $x = (r_e, \theta)$, with $r_e \in [0, 1)$ and $\theta \in [0, 2\pi)$. The distance from the origin, $r_h$ is given by

$$r_h = 2 \operatorname{arctanh} r_e \qquad (1)$$

and the circumference of a circle of hyperbolic radius $R$ is $C = 2\pi \sinh R$. Note that as points approach the edge of the disk, $r_e = 1$, the hyperbolic distance from the origin $r_h$ tends to infinity. In Euclidean neural embeddings, the inner product between vector representations of vertices is used to quantify their similarity. However, unlike Euclidean space, hyperbolic space is not a vector space and there is no global inner product. Instead, given points $\mathbf{x}_1 = (r_1, \theta_1)$ and $\mathbf{x}_2 = (r_2, \theta_2)$ we define a

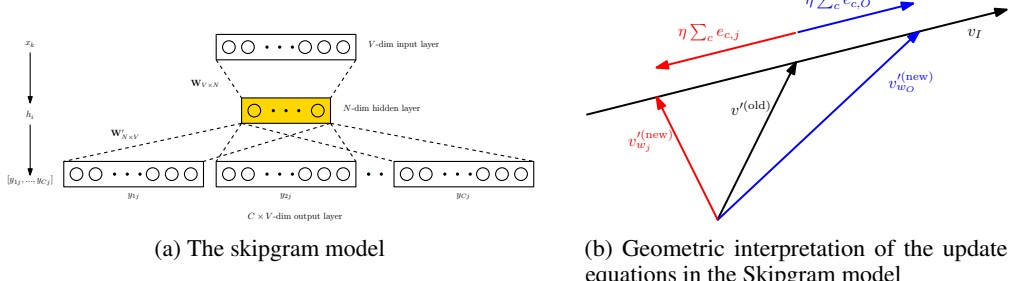

(a) The skipgram model

(b) Geometric interpretation of the update equations in the Skipgram model

Figure 2: a The skipgram architecture. The model predicts the context vertices from a single input vertex. The final embedding is the set of learned weights $\mathbf{W}$. b In the model updates, the vector representation of the context vertex $v_{w_O}'^{(\text{new})}$ is moved closer (blue) to the vector representation of the input vertex $v_I$, while all other vectors $v_{w_j}'^{(\text{new})}$ move further away (red). The magnitude of the change is proportional to the prediction error.

cosine similarity weighted by the hyperbolic distance from the origin as

$$\langle \mathbf{x}_1, \mathbf{x}_2 \rangle_H = \|\mathbf{x_1}\|\|\mathbf{x_2}\| \cos(\theta_1 - \theta_2) = 4 \operatorname{arctanh} r_1 \operatorname{arctanh} r_2 \cos(\theta_1 - \theta_2). \tag{2}$$

It is this function that we will use to quantify the similarity between points in the embedding. We note that using a cosine distance in this way does lose some properties of hyperbolic space such as conformality. Our goal is to learn embeddings that perform well on downstream tasks and the key properties of hyperbolic space that permit this are retained. Trade-offs like this are common in the embeddings literature such as the use of negative sampling (Mnih & Teh, 2012; Mnih, 2013).

## 3 NEURAL EMBEDDING IN HYPERBOLIC SPACE

We adopt the notation of the original Skipgram paper (Mikolov et al., 2013a) whereby the input vertex is $w_I$ and the context / output vertex is $w_O$. The corresponding vector representations are $v_{w_I}$ and $v_{w_O}'$, which are elements of the two vector spaces shown in Fig. 2a, $\mathbf{W}$ and $\mathbf{W}'$ respectively. Skipgram has a geometric interpretation, shown in Fig. 2b for vectors in $\mathbf{W}'$. Updates to $v_{w_j}'$ are performed by simply adding (if $w_j$ is the observed output vertex) or subtracting (otherwise) an error-weighted portion of the input vector. Similar, though slightly more complicated, update rules apply to the vectors in $\mathbf{W}$. Given this interpretation, it is natural to look for alternative geometries that improve on Euclidean geometry.

To embed a graph in hyperbolic space we replace Skipgram's two Euclidean vector spaces ($\mathbf{W}$ and $\mathbf{W}'$ in Fig. 2a) with two Poincaré disks. We learn embeddings by optimising an objective function that predicts context vertices from an input vertex, but we replace the Euclidean dot products used in Skipgram with (2). A softmax function is used for the conditional predictive distribution

$$p(w_O|w_I) = \exp(\langle v_{w_O}', v_{w_I} \rangle_H) / \sum_{i=1}^{V} \exp(\langle v_{w_i}', v_{w_I} \rangle_H), \tag{3}$$

where $v_{w_i}$ is the vector representation of the $i^{th}$ vertex, primed indicates context vectors (see Fig. 2a) and $\langle \cdot, \cdot \rangle_H$ is given in (2). Directly optimising (3) is computationally demanding as the sum in the denominator is over every vertex in the graph. Two commonly used techniques for efficient computation are replacing the softmax with a hierarchical softmax (Mnih & Hinton, 2009; Mikolov et al., 2013a) and negative sampling (Mnih & Teh, 2012; Mnih, 2013). We use negative sampling as it is faster.

### 3.1 MODEL LEARNING

We learn the model using backpropagation with Stochastic Gradient Descent (SGD). Optimisation is conducted in polar native hyperbolic coordinates where $r \in (0, \infty)$, $\theta \in (0, 2\pi]$. For optimisation, this coordinate system has two advantages over the cartesian Euclidean system used by Nickel & Douwe (2017). Firstly there is no need to constrain the optimiser s.t. $\|\mathbf{x}\| < 1$. This is important as arbitrarily moving points a small Euclidean distance inside the disk equates to an infinite hyperbolic

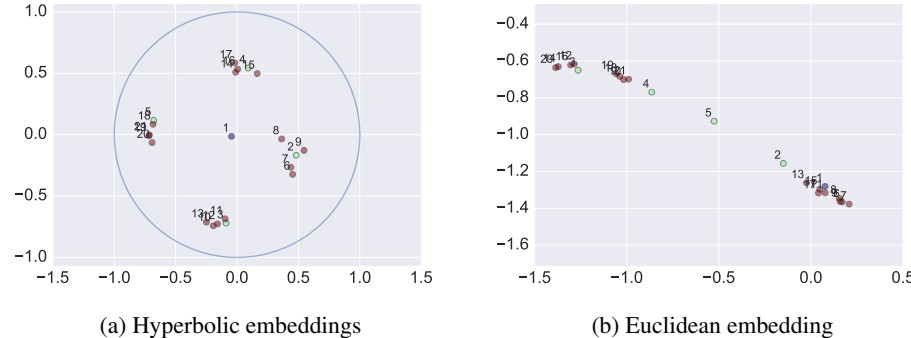

(a) Hyperbolic embeddings           (b) Euclidean embedding

Figure 3: Comparison between the embeddings of a complete 4-ary tree with three levels. Hyperbolic embeddings are able to represent the trees branching factor and position the root at the location of the shortest path length. The Euclidean embedding can not reproduce the isometries of the tree.

distance. Secondly, polar coordinates result in update equations that are simple modifications of the Euclidean updates, which avoids evaluating the metric tensor for each data point. The negative log likelihood using negative sampling is

$$E = -\log \sigma(\langle v'_{w_O}, v_{w_I} \rangle_H) - \sum_{w_j \in W_{neg}} \log \sigma(-\langle v'_{w_j}, v_{w_I} \rangle_H) \tag{4}$$

$$= -\log \sigma(u_O) - \sum_{w_j \in W_{neg}} \mathbb{E}_{w_j \sim P_n}[\log \sigma(-u_j)] \tag{5}$$

where $v_{w_I}$, $v'_{w_O}$ are the vector representation of the input and context vertices, $u_j = \langle v'_{w_j}, v_{w_I} \rangle_H$, $W_{neg}$ is a set of samples drawn from the noise distribution and $\sigma$ is the sigmoid function. The first term represents the observed data and the second term the negative samples. To draw $W_{neg}$, we specify the noise distribution $P_n$ to be the unigram distribution of the vertices in the input sequence raised to $3/4$ as in (Mikolov et al., 2013a). The gradient of the negative log-likelihood in (5) w.r.t. $u_j$ is given by

$$\frac{\partial E}{\partial u_j} = \epsilon_j = \begin{cases} \sigma(u_j) - 1, & \text{if } w_j = w_O \\ \sigma(u_j), & \text{if } w_j \in W_{neg} \\ 0, & \text{otherwise} \end{cases} . \tag{6}$$

The derivatives w.r.t. the components of vectors in $\mathbf{W}'$ (in natural polar hyperbolic coordinates) are

$$\frac{\partial E}{\partial (r'_j)_k} = \frac{\partial E}{\partial u_j} \frac{\partial u_j}{\partial (r'_j)_k} = \frac{\partial E}{\partial u_j} r_I \cos(\theta_I - \theta'_j), \qquad \frac{\partial E}{\partial (\theta'_j)_k} = \frac{\partial E}{\partial u_j} r'_j r_I \sin(\theta_I - \theta'_j), \tag{7}$$

such that the Jacobian is $\nabla_{\mathbf{r}} E = \frac{\partial E}{\partial r} \hat{\mathbf{r}} + \frac{1}{\sinh r} \frac{\partial E}{\partial \theta} \hat{\theta}$. This leads to

$$r_j'^{new} = \begin{cases} r_j'^{old} - \eta \epsilon_j r_I \cos(\theta_I - \theta'_j), & \text{if } w_j \in \chi \\ r_j'^{old}, & \text{otherwise} \end{cases} \tag{8}$$

$$\theta_j'^{new} = \begin{cases} \theta_j'^{old} - \eta \epsilon_j \frac{r_I r_j}{\sinh r_j} \sin(\theta_I - \theta'_j), & \text{if } w_j \in \chi \\ \theta_j'^{old}, & \text{otherwise} \end{cases} \tag{9}$$

where $\chi = w_O \cup W_{neg}$, $\eta$ is the learning rate and $\epsilon_j$ is the prediction error defined in (6). Calculating the derivatives w.r.t. the input embedding follows the same pattern, and we obtain

$$\frac{\partial E}{\partial r_I} = \sum_{j:w_j \in \chi} \frac{\partial E}{\partial u_j} r'_j \cos(\theta_I - \theta'_j), \qquad \frac{\partial E}{\partial \theta_I} = \sum_{j:w_j \in \chi} -\frac{\partial E}{\partial u_j} r_I r'_j \sin(\theta_I - \theta'_j). \tag{10}$$

The corresponding update equations are

$$r_I^{new} = r_I^{old} - \eta \sum_{j:w_j \in \chi} \epsilon_j r'_j \cos(\theta_I - \theta'_j), \qquad \theta_I^{new} = \theta_I^{old} - \eta \sum_{j:w_j \in \chi} \epsilon_j \frac{r_I r'_j}{\sinh r_I} \sin(\theta_I - \theta'_j), \tag{11}$$

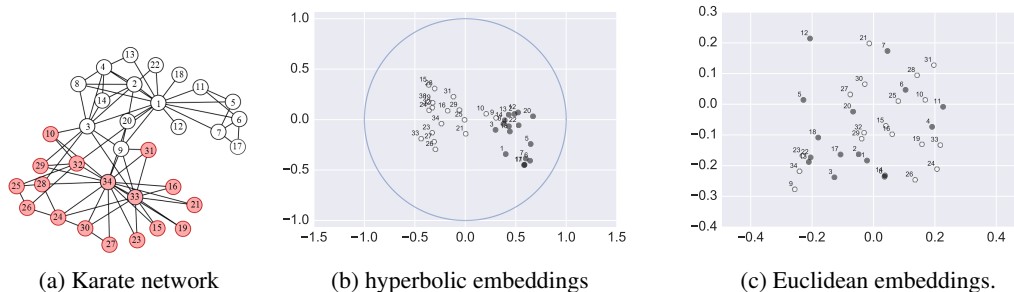

(a) Karate network  (b) hyperbolic embeddings  (c) Euclidean embeddings.

Figure 4: The two factions of the Zachary karate network are linearly separable when embedded in 2D hyperbolic, but not Euclidean space. Both embeddings were run for 5 epochs on the same intermediate random walks.

where $t_j$ is an indicator variable s.t. $t_j = 1$ iff $w_j = w_O$, and $t_j = 0$ otherwise. Following optimisation, the vectors are mapped back to Euclidean coordinates on the Poincaré disk through $\theta_h \to \theta_e$ and $r_h \to \tanh \frac{r_h}{2}$. The asymptotic runtimes of the update equations (8)–(9) and (11) are the same as Euclidean Skipgram, i.e., the hyperbolic embedding does not add computational burden.

## 4 EXPERIMENTAL EVALUATION

In this section, we assess the quality of hyperbolic embeddings and compare them to embeddings in Euclidean spaces. Firstly we perform a qualitative assessment of the embeddings on a synthetic fully connected tree graph and a small social network. It is clear that embeddings in hyperbolic space exhibit a number of features that are superior to Euclidean embeddings. Secondly we run experiments on a number of public benchmark networks, producing both Euclidean and hyperbolic embeddings and contrasting the performance of both on a downstream vertex classification task. We provide a TensorFlow implementation and datasets to replicate our experiments in our github repository [1].

### 4.1 QUALITATIVE ASSESSMENT

To illustrate the usefulness of hyperbolic embeddings we visually compare hyperbolic embeddings with Euclidean plots. In all cases, embeddings were generated using five training epochs on an intermediate dataset of ten-step random walks, one originating at each vertex. Figures 3 and 4 show hyperbolic embeddings in the 2D Poincaré model of hyperbolic space where the circles of radius 1 is the infinite boundary and Euclidean embeddings in $\mathbb{R}^2$. Fig. 3 shows embeddings of a complete 4-ary tree with three levels. The vertex numbering is breadth first with one for the root and 2, 3, 4, 5 for the second level etc. The hyperbolic embedding has the root vertex close to the origin of the disk, which is the position with the shortest average path length. The leaves are all located in close proximity to their parents, and there are clearly four clusters representing the tree's branching factor. The Euclidean embedding is incapable of representing the tree structure with adjacent vertices at large distances (such as 1 and 3) and vertices that are maximally separated in the tree appearing close in the embedding (such as 19 and 7).

Fig. 4a shows the 34-vertex karate network, which is split into two factions. Fig. 4b shows the hyperbolic embedding of this network where the two factions can be clearly separated. In addition, the vertices (5, 6, 7, 11, 17) in Fig. 4a are the junior instructors, who are forbidden by the instructor (vertex 1) from socialising with other members of the karate club. For this reason they form a community that is only connected through the instructor. This community is clearly visible in Fig. 4b to the

Table 1: Experimental datasets. 'Largest class' gives the fraction of the dataset composed by the largest class and thereby provides the benchmark for random prediction accuracy.

| name | $|V|$ | $|E|$ | $|y|$ | largest class | Labels |
|---|---|---|---|---|---|
| karate | 34 | 77 | 2 | 0.53 | Factions |
| polbooks | 105 | 441 | 3 | 0.46 | Affiliation |
| football | 115 | 613 | 12 | 0.11 | League |
| adjnoun | 112 | 425 | 2 | 0.52 | Part of Speech |
| polblogs | 1,224 | 16,781 | 2 | 0.52 | Affiliation |

---

[1]https://github.com/anonymous/authors

Table 2: Average F1 scores across experiments. HB is HyBed, P is Poincaré and D indicates a Deepwalk embedding. 'All' gives a global average across the five experiments.

| experiment | HB | P | D2 | D4 | D8 | D16 | D32 | D64 | D128 |
|---|---|---|---|---|---|---|---|---|---|
| karate | **0.91** | 0.90 | 0.37 | 0.37 | 0.36 | 0.36 | 0.37 | 0.37 | 0.38 |
| polbooks | **0.62** | 0.59 | 0.27 | 0.24 | 0.21 | 0.21 | 0.21 | 0.21 | 0.21 |
| football | **0.22** | 0.14 | 0.03 | 0.03 | 0.04 | 0.04 | 0.04 | 0.04 | 0.04 |
| adjnoun | 0.49 | 0.47 | **0.53** | 0.50 | 0.44 | 0.38 | 0.36 | 0.36 | 0.36 |
| polblogs | 0.90 | **0.91** | 0.63 | 0.64 | 0.67 | 0.64 | 0.63 | 0.55 | 0.51 |
| All | **0.63** | 0.60 | 0.37 | 0.35 | 0.34 | 0.33 | 0.32 | 0.30 | 0.30 |

right of the graph. The Euclidean embedding in Fig. 4c fails to capture these important features of the underlying graph.

## 4.2 Vertex Attribute Prediction

We quantitatively evaluate the success of neural embeddings in hyperbolic space by using the learned embeddings to predict held-out labels of vertices in networks. In our experiments, we compare our embedding to Euclidean embeddings of dimensions 2, 4, 8, 16, 32, 64 and 128 and the hyperbolic embeddings of Nickel & Douwe (2017). To generate embeddings we first create an intermediate dataset by taking a series of random walks over the networks. For each network we use a set of ten-step random walks with one walk originating at each vertex.

The embeddings are all trained using the same parameters and intermediate random walk dataset. For Euclidean embeddings we use the gensim (Rehurek & Sojka, 2010) python package, while the HyBed embeddings and our implementation of Nickel & Douwe (2017) are written in TensorFlow. In both cases, we use five training epochs, a context width of five (giving 10 context vertices per input vertex) and a linearly decaying SGD optimiser with initial learning rate of 0.2. We do not prune any vertices.

We report results on five publicly available network datasets for the problem of vertex attribution. 1. Karate: Zachary's karate club contains 34 vertices divided into two factions (Zachary, 1977); 2. Polbooks: A network of books about US politics published around the time of the 2004 presidential election and sold by the online bookseller Amazon.com. Edges between books represent frequent co-purchasing of books by the same buyers; 3. Football: A network of American football games between Division IA colleges during regular season Fall 2000 (Girvan & Newman, 2002); 4. Adjnoun: Adjacency network of common adjectives and nouns in the novel David Copperfield by Charles Dickens (Newman, 2006); 5. Polblogs: A network of hyperlinks between weblogs on US politics, recorded in 2005 (Adamic & Glance, 2005). Statistics for these datasets are recorded in Table 1.

The results of our experiments together with the HyBed and 2D Deepwalk embeddings used to derive them are shown in Fig. 5. The vertex colours of the embedding plots indicate different values of the vertex labels. The legend shown in Fig. 5a applies to all line graphs. The line graphs show macro F1 scores against the percentage of labelled data used to train a logistic regression classifier with the embeddings as features. Here we follow the method for generating multi-label F1 scores described in (Liu et al., 2006). The error bars show one standard error from the mean over ten repetitions. The blue lines show HyBed hyperbolic embeddings, the yellow lines give the 2D Poincaré embeddings of Nickel & Douwe (2017) while the red lines depict Deepwalk embeddings at various dimensions.

As we use one-vs-all logistic regression with embedding coordinates as features, good embeddings are those that can linearly separate one class from all other classes. Figure 5 shows that HyBed embeddings tend to cluster together similar classes so that they are linearly separable from other classes, unlike the Euclidean embeddings.

Table 2 gives the average F1 score for each experiment. The hyperbolic methods are greatly superior to Euclidean methods and HyBed is best in three of the five experiments with the highest global average F1 score. The only experiment in which the performance of Euclidean embeddings is comparable with hyperbolic methods is the Word Adjacency dataset and in this case Figures 5j shows that the error bars are wide and overlapping.

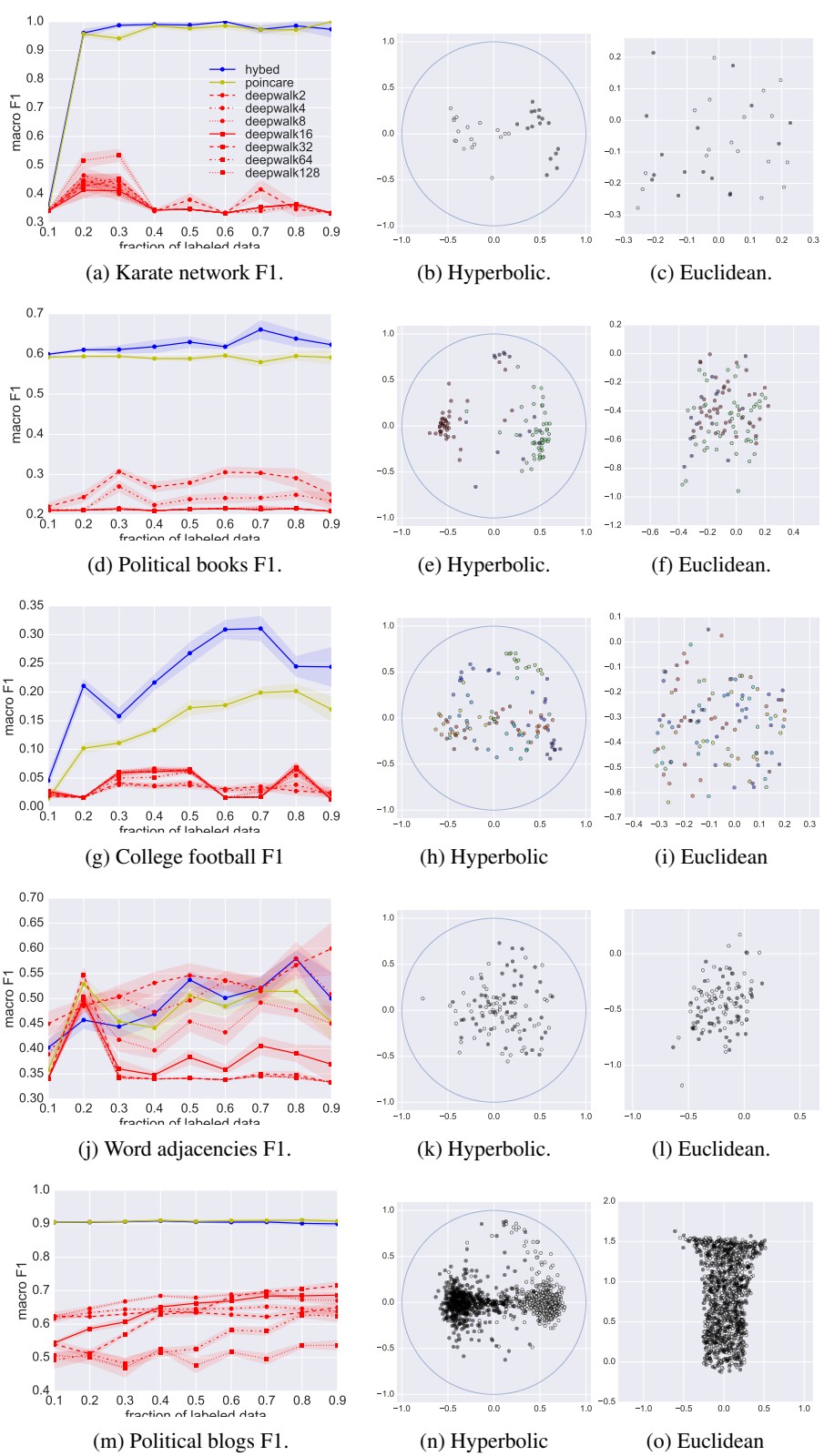

Figure 5: Each row contains a line plot of macro F1 score for predicting held-out vertex labels from embedded representations using logistic regression together with the HyBed and 2D Euclidean embeddings. Error bars are standard errors from the mean over ten repetitions.

## 5 CONCLUSION

We have introduced the concept of neural embeddings in hyperbolic space. Our model is based on the Skipgram with negative sampling architecture and optimises embeddings in native spherical hyperbolic coordinates. Hyperbolic space has the property that power-law degree distributions, strong clustering and hierarchical community structure emerge naturally when random graphs are embedded in hyperbolic space. It is therefore logical to exploit the structure of the hyperbolic space for useful embeddings of complex networks. We have demonstrated that when applied to the task of classifying vertices of complex networks, hyperbolic space embeddings significantly outperform embeddings in Euclidean space. Furthermore, the neural hyperbolic approach we develop here performs well when compared to the most comparable method, that of Poincaré embeddings.

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
