# OpenReview forum: "Hybed: Hyperbolic Neural Graph Embedding"
_ICLR.cc/2018/Conference — Reject_

### Official Review · AnonReviewer2 · 2017-11-20
**Done before?**

**Rating:** 4
**Confidence:** 3

**Review:**

== Preamble ==

As promised, I have read the updated paper from scratch and this is my revised review. My original review is kept below for reference. My original review had rating "4: Ok but not good enough - rejection".

== Updated review ==

The revised improves upon the original submission in several ways and, in particular, does a much better job at positioning itself within the existing body of literature. The new experiments also indicate that the proposed model offer some improvement over Nickel & Douwe, NIPS 2017).

I do have remaining concerns that unfortunately still prevent me from recommending acceptance:

- Throughout the paper it is argued that we should embed into a hyperbolic space. Such a space is characterized by its metric, but the proposed model do not use a hyperbolic metric. Rather it relies on a heuristic similarity measure that is inspired by the hyperbolic metric. I understand that this may be a practical choice, but then I find it misleading that the paper repeatedly states that points are embedded into a hyperbolic space (which is incorrect). This concern was also raised on this forum prior to the revision.

- The resulting optimization is one of the key selling points of the proposed method as it is unconstrained while Nickel & Douwe resort to a constrained optimization. Clearly unconstrained optimization is to be preferred. However, it is not entirely correct (from what I understand), that the resulting optimization is indeed unconstrained. Nickel & Douwe work under the constraint that |x| < 1, while the proposed model use polar coordinates (r, theta): r in (0, infinity) and theta in (0, 2 pi]. Note that theta parametrize a circle, and therefore wrapping may occur (this should really be mentioned in the paper). The constraints on theta are quite easy to cope with, so I agree with the authors that they have a more simple optimization problem. However, this is only true since points are embedded on the unit disk (2D). Should you want to embed into higher dimensional spaces, then theta need to be confined to live on the unit sphere, i.e. |theta| = 1 (the current setting is just a special-case of the unit sphere). While optimizing over the unit sphere is manageable it is most definitely a constrained optimization problem, and it is far from clear that it is much easier than working under the Nickel & Douwe constraint, |x| < 1.

Other comments:
- The sentence "even infinite trees have nearly isometric embeddings in hyperbolic space (Gromov, 2007)" sounds cool (I mean, we all want to cite Gromov), but what does it really mean? An isometric embedding is merely one that preserves a metric, so this statement only makes sense if the space of infinite trees had a single meaningful metric in the first place (it doesn't; that's a design choice).

- In the "Contribution" and "Conclusion" sections it is claimed that the paper "introduce the new concept of neural embeddings in hyperbolic space". I thought that was what Nickel & Douwe did... I understand that the authors are frustrated by this parallel work, but at this stage, I don't think the present paper can make this "introducing" claim.

- The caption in Figure 2 miss some indication that "a" and "b" refer to subfigures. I recommend "a" --> "a)" and "b" --> "b)".

- On page 4 it is mentioned that under the heuristic similarity measure some properties of hyperbolic spaces are lost while other are retained. From what I can read, it is only claimed that key properties are kept; a more formal argument (even if trivial) would have been helpful.


== Original Review ==

The paper considers embeddings of graph-structured data onto the hyperbolic Poincare ball. Focus is on word2vec style models but with hyperbolic embeddings. I am unable to determine how suitable an embedding space the Poincare ball really is, since I am not familiar enough with the type of data studied in the paper. I have a few minor comments/questions to the work, but my main concern is a seeming lack of novelty:
The paper argues that the main contribution is that this is the first neural embedding onto a hyperbolic space. From what I can see, the paper

  Poincaré Embeddings for Learning Hierarchical Representations
  https://arxiv.org/abs/1705.08039

consider an almost identical model to the one proposed here with an almost identical motivation and application set. Some technicalities appear different, but (to me) it seems like the main claimed novelties of the present paper has already been out for a while. If this analysis is incorrect, then I encourage the authors to provide very explicit arguments for this in the rebuttal phase.

Other comments:
*) It seems to me that, by construction, most data will be pushed towards the boundary of the Poincare ball during the embedding. Is that a property you want?
*) I found it rather surprising that the log-likelihood under consideration was pushed to an appendix of the paper, while its various derivatives are part of the main text. Given the not-so-tight page limits of ICLR, I'd recommend to provide the log-likelihood as part of the main text (it's rather difficult to evaluate the correctness of a derivative when its base function is not stated).
*) In the introduction must energy is used on the importance of large data sets, but it appears that only fairly small-scale experiments are considered. I'd recommend a better synchronization.
*) I find visual comparisons difficult on the Poincare ball as I am so trained at assuming Euclidean distances when making visual comparisons (I suspect most readers are as well). I think one needs to be very careful when making visual comparisons under non-trivial metrics.
*) In the final experiment, a logistic regressor is fitted post hoc to the embedded points. Why not directly optimize a hyperbolic classifier?

Pros:
+ well-written and (fairly) well-motivated.

Cons:
- It appears that novelty is very limited as highly similar work (see above) has been out for a while.

---

> ### Author Response · Authors · 2017-12-07
> **Both papers were written at the same time and the approaches differ**
>
> Our paper is similar to “Poincaré Embeddings for Learning Hierarchical Representations” (Nickel and Kiela, NIPS 2017). However, both were written independently at the same time. While they are similar, there are several important differences between them. Our work uses a spherical coordinate system, instead of cartesian coordinates, which we believe is more elegant as it exploits the symmetry of the Poincare ball. In addition, we use natural hyperbolic coordinates that extend radially (0,inf) instead of (0,1). Natural coordinates remove the numerical issues at the boundary of the ball. In the paper by Nickel and Kiela, this is a problem as the optimization will push points to values greater than 1. To fix this, they introduce a heuristic that moves points a small distance back inside the ball when the optimizer pushes them outside. As most of the space is towards the edge of the ball, many of the points in their system are effectively placed solely by the heuristic. This is not a problem in natural coordinates and we simply switch back coordinate systems on completion of the optimization process. In addition, we use a different similarity metric based on cosine similarity instead of the hyperbolic distance.
>
> Despite the similarity, it is worthwhile publishing this paper as it provides a complementary perspective on the problem and grows the evidence base for this as a powerful technique that other researchers can employ. Our paper shows that the principle of hyperbolic embedding can be achieved using a cosine-similarity approach as well as the distance-similarity approach used in the other paper, and so demonstrates a more general applicability of the idea. An example of where this has previously occurred are Auto-Encoding Variational Bayes (Kingma and Welling, ICLR 2014) https://arxiv.org/abs/1312.6114 and later Stochastic Backpropagation and Approximate Inference in Deep Generative Models Rezende et al., ICML 2014) https://arxiv.org/abs/1401.4082
> Both papers contained the same idea and appeared on arxiv within a month, but provided different perspectives.

---

> > ### Comment · AnonReviewer2 · 2017-12-08
> > **I'm sympathetic, but I need stronger arguments**
> >
> > I'm sympathetic to your position; on a personal level I understand the frustration of seeing a large body of work reduced simply because somebody else got there infinitesimally faster.
> >
> > With that in mind, I'm willing to listen to the arguments of why this paper brings "new perspective" to the table. You bring up two key differences:
> >
> > 1) You use a natural parametrization of the Poincare ball, which you state is more "elegant". I see that there is some benefit to having an unconstrained optimization vs one which is constrained. Is that what you mean by "elegant" or is there more to this claimed elegance? Can you empirically quantify the benefits of your parametrization?
> >
> > 2) You use a cosine-based similarity measure rather than the standard hyperbolic distance. You mention this is a difference, but does it also offer benefits? From a geometric point of view, I guess it seems more "elegant" to use the standard hyperbolic distance.

---

> > > ### Author Response · Authors · 2017-12-13
> > > **It allows for a more efficient better behaved optimizer**
> > >
> > > Thank you, we appreciate your sympathy.
> > >
> > > The Poincare representation has circular symmetry such that all elements on the boundary of a circle centered at the origin have the same distortion to the Euclidean length. By representing the system in spherical co-ordinates the update equations are greatly simplified with the radial updates being exactly the Euclidean updates and the angular update a simple modification of the Euclidean. This simplicity could explain why we are able to get compelling results after just 5 epochs while “Poincaré Embeddings for Learning Hierarchical Representations” (Nickel and Kiela, NIPS 2017) require a 20 epoch burn in period.
> > >
> > > In addition, the heuristic of  “Poincaré Embeddings for Learning Hierarchical Representations” (Nickel and Kiela, NIPS 2017) that places points that the optimizer pushes outside of the boundary of the Poincare ball is troubling as it requires arbitrarily moving some  points an infinite hyperbolic distance, which seems like an undesirable property for an optimizer. This problem does not exist in natural coordinates as the Poincare ball boundary is at infinity.

---

> > > > ### Comment · AnonReviewer2 · 2017-12-14
> > > > **Let's see it in writing**
> > > >
> > > > Okay, I think that's a reasonable point. From this description, I'm having a hard time determining if this is an important or a minor insight, but it sounds like there's a valuable insight.
> > > >
> > > > But I'll make you an offer: if you update the paper to discuss these matters in detail, then I'll re-review the paper from scratch. Please also take comments from the other reviewers into account. If you can document that there's an empirical benefit to your parametrization (you already hinted at that), then that's also great (given the short time available, I acknowledge that this may not be possible).

---

### Official Review · AnonReviewer3 · 2017-11-21
**nice paper, good concepts, not enough experiments.**

**Rating:** 7
**Confidence:** 2

**Review:**

This paper proposes tree vertex embeddings over hyperbolic space. The conditional predictive distribution is the softmax of <v1, v2>_H = ||v1|| ||v2|| cos(theta1-theta2), and v1, v2 are points  defined via polar coordinates (r1,theta1), and (r2,theta2).
To evaluate, the authors show some qualitative embeddings of graph and 2-d projections, as well as F1 scores in identifying the biggest cluster associated with a class.

The paper is well motivated, with an explanation of the technique as well as its applications in tree embedding in general. I also like the evaluations, and shows a clear benefit of this poincare embedding vs euclidean embedding.

However, graph embeddings are now a very well explored space, and this paper does not seem to mention or compare against other hyperbolic (or any noneuclidean) embedding techniques. From a 2 second google search, I found several sources with very similar sounding concepts:

Maximilian Nickel, Douwe Kiela, Poincaré Embeddings for Learning Hierarchical Representations

A Cvetkovski, M Crovella, Hyperbolic Embedding and Routing for Dynamic Graphs

Yuval Shavitt, Tomar Tankel, Hyperbolic Embedding of Internet Graph for Distance Estimation and Overlay Construction

Thomas Bläsius, Tobias Friedrich, Anton Krohmer, andSören Laue. Efficient Embedding of Scale-Free Graphs in the Hyperbolic Plane

I think this paper does have some novelty in applying it to the skip-gram model and using deep walk, but it should make more clear that using hyperbolic space embeddings for graphs is a popular and by now, intuitive construct. Along the same lines, the benefit of using the skip-gram and deep-walk techniques should be compared against some of the other graph embedding techniques out there, of which none are listed in the experiment section.

Overall, a detailed comparison against 1 or 2 other hyperbolic graph embedding techniques would be sufficient for me to change my vote to accept.

---

> ### Author Response · Authors · 2017-12-13
> **Thank you for your helpful comments**
>
> Thank you for your helpful comments. We will add comparisons with more graph embedding methods and expand the literature review in future versions.

---

> > ### Comment · AnonReviewer3 · 2018-01-07
> > **reply to revision**
> >
> > The authors have addressed my major issue so I have changed my vote to accept.

---

### Official Review · AnonReviewer1 · 2017-12-11
**Nice idea but missing literature, poor paper organisation, and experiments that could be more complete**

**Rating:** 5
**Confidence:** 3

**Review:**

The authors present a neural embedding technique using a hyperbolic space.
The idea of embedding data into a space that is not Euclidean is not new.
There have been attempts to project onto (hyper)spheres.
Also, the proposal bears some resemblance with what is done in t-SNE, where an (exponential) distortion of distances is induced. Discussing this potential similarity would certainly broaden the readership of the paper.

The organisation of the paper might be improved, with a clearer red line and fewer digressions.
The call to the very small appendix via eq. 17 is an example.
The position of Table in the paper is odd as well.
The order of examples in Fig.5 differs from the order in the list.

The experiments are well illustrative but rather small sized.
The qualitative assessment is always interesting and it is completed with some label prediction task.
Due the geometrical consideretations developed in the paper, other quality criteria like e.g. how well neighbourhoods are preserved in the embeddings would give some more insights.

All in all the idea developed in the paper sounds interesting but the paper organisation seems a bit loose and additional aspects should be investigated.

---

> ### Author Response · Authors · 2017-12-13
> **Thank you for your suggestions on how to improve the paper**
>
> Thank you for your suggestions on how to improve the paper. We do not claim that the non-Euclidean embeddings are new, only that doing so through a neural network is and that a change in geometry can markedly improve the representations learned by the very popular SkipGram / Word2vec model. We agree that eq. 17 should be moved to the main body and that it would improve readability to re-order some of the tables.

---

### Official Review · AnonReviewer4 · 2017-12-12
**Concerns about novelty**

**Rating:** 4
**Confidence:** 3

**Review:**

The authors present a method to embed graphs in hyperbolic space, and show that this approach yields stronger attribute predictions on a set of graph datasets. I am concerned by the strong similarity between this work and Poincaré Embeddings for Learning Hierarchical Representations (https://arxiv.org/abs/1705.08039). The latter has been public since May of this year, which leads me to doubt the novelty of this work.

I also find the organization of the paper to be poor.
- There is a surprisingly high number of digressions.
- For some reason, Eq 17 is not included in the main paper. I would argue that this equation is one of the most important equations in the paper, given that it is the one you are optimizing.
- The font size in the main result figure is so small that one cannot hope to parse what the plots are illustrating.
- I am not sure what insights the readers are supposed to gain from the visual comparisons between the Euclidean and Poincare embeddings.

Due to the poor presentation, I actually have difficulty making sense of the evaluation in this paper (it would help if the text was legible). I think this paper requires significant work and it not suitable for publication in its current state.

As a kind of unrelated note. It occurs to me that papers on hyperbolic embeddings tend to evaluate evaluate on attribute or link prediction. It would be great if authors would also evaluate these pretrained embeddings on downstream applications such as relation extraction, knowledge base population etc.

---

> ### Author Response · Authors · 2017-12-13
> **Both papers were written at the same time and the approaches differ**
>
> Our paper is similar to “Poincaré Embeddings for Learning Hierarchical Representations” (Nickel and Kiela, NIPS 2017). However, both were written independently at the same time. While they are similar, there are several important differences between them. Our work uses a spherical coordinate system, instead of cartesian coordinates, which we believe is more elegant as it exploits the symmetry of the Poincare ball. In addition, we use natural hyperbolic coordinates that extend radially (0,inf) instead of (0,1). Natural coordinates remove the numerical issues at the boundary of the ball. In the paper by Nickel and Kiela, this is a problem as the optimization will push points to values greater than 1. To fix this, they introduce a heuristic that moves points a small distance back inside the ball when the optimizer pushes them outside. As most of the space is towards the edge of the ball, many of the points in their system are effectively placed solely by the heuristic. This is not a problem in natural coordinates and we simply switch back coordinate systems on completion of the optimization process. In addition, we use a different similarity metric based on cosine similarity instead of the hyperbolic distance.
>
> Despite the similarity, it is worthwhile publishing this paper as it provides a complementary perspective on the problem and grows the evidence base for this as a powerful technique that other researchers can employ. Our paper shows that the principle of hyperbolic embedding can be achieved using a cosine-similarity approach as well as the distance-similarity approach used in the other paper, and so demonstrates a more general applicability of the idea. An example of where this has previously occurred are Auto-Encoding Variational Bayes (Kingma and Welling, ICLR 2014) https://arxiv.org/abs/1312.6114 and later Stochastic Backpropagation and Approximate Inference in Deep Generative Models Rezende et al., ICML 2014) https://arxiv.org/abs/1401.4082
> Both papers contained the same idea and appeared on arxiv within a month, but provided different perspectives.
>
> In response to the other comments, eq. 17 was removed as it is well known, but we agree that this affects readability and will include this in the main text of future versions. Similarly we will increase the font size of the axis labels. The comparisons of hyperbolic and Euclidean embeddings are showing that in hyperbolic space the classes are more easily separable. This is important as the results use a logistic regression classifier.

---

### Public Comment · (anonymous) · 2017-11-22
**These embeddings are not in the hyperbolic space as claimed...**

This paper is completely wrong: by changing the dot-product, you cannot talk about a hyperbolic space anymore.

The dot-product given by the authors has nothing to do with the hyperbolic riemannian metric.

---

> ### Author Response · Authors · 2017-11-24
> **We define a similarity in the objective function for points in hyperbolic space. We do not claim that this is a dot product.**
>
>
> Hyperbolic space is not a vector space and so does not have a globally defined inner-product. We have defined a measure of similarity between embedded points, which is a cosine similarity weighted by the distance in hyperbolic space.
> The hyperbolic metric comes into this because it is the hyperbolic distances from the origin (using this metric) that weight the cosine distance.
> The net effect is that when the coordinates of points are updated, the updates in angular directions (ie. perpendicular to the radial direction) are suppressed for points far away from the origin, by a factor related to their distance from the origin in the hyperbolic space. This has the desired effect of allowing many peripheral points to be mutually distant, while simultaneously close to central points, as, for example, in figures 3 and 4.
> The intention is to be able to use the machinery of neural embeddings while also getting the useful geometric properties of hyperbolic space.

---

> > ### Public Comment · (anonymous) · 2017-11-26
> > **The similarity function still doesn't reflect the hyperbolic geometry.**
> >
> > What is misleading is to describe these embeddings as inheriting the geometrical properties of the hyperbolic space, which isn't true.
> >
> > It is true that the similarity function defined by the authors inherits some of the properties of the hyperbolic space, as they explained it above.
> >
> > However, it should be emphasized that it is only vaguely related to the hyperbolic metric via some heuristic, and that most properties characterizing a space endorsed with a hyperbolic structure will not be satisfied by the word-embedding space.
> >
> > Namely, with this similarity measure, one loses the possibility to use the conformality of the hyperbolic metric, which gave us closed forms to compute curvature tensors, volume elements, the metric tensor, the exponential map and geodesics... Which should be clearly stated at the beginning of the paper, in order to not mislead readers expecting a real exploitation of hyperbolic geometry in the embedding space.

---

> > > ### Author Response · Authors · 2017-11-30
> > > **We are mostly concerned with learning high quality embeddings**
> > >
> > > We agree with the comment that the similarity is somewhat heuristic and inspired by the hyperbolic geometry. We care about the quality of the embeddings most of all. The use of heuristics to find good embeddings is quite common in the literature. For instance, negative sampling is often used, which abandons the strict maximization of the log probability of the softmax to learn more efficiently. What we are doing is in the same spirit as negative sampling.

---

> ### Public Comment · (anonymous) · 2017-12-12
> **Completely agree - These are not hyperbolic embeddings**
>
> Since hyperbolic space is not a vector space (only a metric space) it doesn't make sense to define an inner product between two points on the manifold. This is a significant flaw in the paper.

---

### Comment · AnonReviewer1 · 2017-12-11
**The paper somehow overlooks literature from dimensionality reduction**

The authors present a neural embedding technique using a hyperbolic space.
The idea of embedding data into a space that is not Euclidean is not new.
There have been attempts to project onto (hyper)spheres.
Also, the proposal bears some resemblance with what is done in t-SNE, where an (exponential) distortion of distances is induced. Discussing this potential similarity would certainly broaden the readership of the paper.

The organisation of the paper might be improved, with a clearer red line and fewer digressions.
The call to the very small appendix via eq. 17 is an example.
The position of Table in the paper is odd as well.
The order of examples in Fig.5 differs from the order in the list.

The experiments are well illustrative but rather small sized.
The qualitative assessment is always interesting and it is completed with some label prediction task.
Due the geometrical consideretations developed in the paper, other quality criteria like e.g. how well neighbourhoods are preserved in the embeddings would give some more insights.

All in all the idea developed in the paper sounds interesting but the paper organisation seems a bit loose and additional aspects should be investigated.

---

> ### Author Response · Authors · 2017-12-13
> **This seems to be a duplicate comment**
>
> This seems to be a duplicate comment

---

### Author Response · Authors · 2018-01-05
**Revised version addressing the reviewers comments**

We thank the reviewers for their careful consideration of our paper and for offering improvements to the manuscript. We have submitted a new version of the paper that addresses the comments of the reviewers:

Related work on hyperbolic embeddings of graphs has been added. The paper ‘Poincaré Embeddings for Learning Hierarchical Representations’, which appeared concurrently with ours and was mentioned by several reviewers is now explicitly referenced and the differences between this work and our own are described and quantitatively evaluated.

An explanation of how our embedding sacrifices some of the properties of the hyperbolic space and why this is acceptable in the context of learning embeddings has been added.

The appendix has been merged with the main text to include Eq 17.  as several reviewers identified that this inhibited readability.

The results section has been improved by including a comparison with the hyperbolic method of ‘Poincaré Embeddings for Learning Hierarchical Representations’ and the charts have been regenerated with larger labels and displayed in an order that is consistent with the tables. A new results table has been added to ease quantitative comparisons between methods. An explanation of how the embedding diagrams in Fig 5 should be interpreted has also been added.

---

### Decision · Program_Chairs · 2018-01-29
**ICLR 2018 Conference Acceptance Decision**

**Decision:**

Reject

**Comment:**

This paper does not meet the acceptance bar this year, and thus I must recommend it for rejection.